# Data-Independent Neural Pruning via Coresets

**Ben Mussay**
Computer Science Department
University of Haifa
Haifa, Israel
bengordoncshaifa@gmail.com

**Margarita Osadchy**
Computer Science Department
University of Haifa
Haifa, Israel
rita@cs.haifa.ac.il

**Vladimir Braverman**
Computer Science Department
Johns Hopkins University
Baltimore, MD., USA
vova@cs.jhu.edu

**Samson Zhou**
Computer Science Department
Carnegie Mellon University
Pittsburgh, IN., USA
samsonzhou@gmail.com

**Dan Feldman**
Computer Science Department
University of Haifa
Haifa, Israel
dannyf.post@gmail.com

## Abstract

Previous work showed empirically that large neural networks can be significantly reduced in size while preserving their accuracy. Model compression became a central research topic, as it is crucial for deployment of neural networks on devices with limited computational and memory resources. The majority of the compression methods are based on heuristics and offer no worst-case guarantees on the trade-off between the compression rate and the approximation error for an arbitrarily new sample.

We propose the first efficient, data-independent neural pruning algorithm with a provable trade-off between its compression rate and the approximation error for any future test sample. Our method is based on the coreset framework, which finds a small weighted subset of points that provably approximates the original inputs. Specifically, we approximate the output of a layer of neurons by a coreset of neurons in the previous layer and discard the rest. We apply this framework in a layer-by-layer fashion from the top to the bottom. Unlike previous works, our coreset is data independent, meaning that it provably guarantees the accuracy of the function for any input $x \in \mathbb{R}^d$, including an adversarial one. We demonstrate the effectiveness of our method on popular network architectures. In particular, our coresets yield 90% compression of the LeNet-300-100 architecture on MNIST while improving classification accuracy.

## 1 Introduction

Neural networks today are the most popular and effective instrument of machine learning with numerous applications in different domains. Since Krizhevsky et al. (2012) used a model with 60M parameters to win the ImageNet competition in 2012, network architectures have been growing wider and deeper. The vast overparametrization of neural networks offers better convergence (Allen-Zhu et al., 2019) and better generalization (Neyshabur et al., 2018). The downside of the overparametrization is its high memory and computational costs, which prevent the use of these networks in small devices, e.g., smartphones. Fortunately, it was observed that a trained network could be reduced to smaller sizes without much accuracy loss. Following this observation, many approaches to

compress existing models have been proposed (see Gale et al. (2019) for a recent review on network sparsification, and Mozer & Smolensky (1989); Srivastava et al. (2014); Yu et al. (2018); He et al. (2017) for neural pruning).

Although a variety of model compression heuristics have been successfully applied to different neural network models, such as Jacob et al. (2018); Han et al. (2015); Alvarez & Salzmann (2017), these approaches generally lack strong provable guarantees on the **trade-off between the compression rate and the approximation error**. The absence of worst-case performance analysis can potentially be a glaring problem depending on the application. Moreover, data-dependent methods for model compression (e.g., Mozer & Smolensky (1989); Srivastava et al. (2014); Hu et al. (2016); Yu et al. (2018); Baykal et al. (2018)) rely on the statistics presented in a data set. Hence, these methods are vulnerable to adversarial attacks (Szegedy et al., 2014), which design inputs that do not follow these statistics.

Ideally, a network compression framework should 1) provide provable guarantees on the trade-off between the compression rate and the approximation error, 2) be data independent, 3) provide high compression rate, and 4) be computationally efficient. To address these goals, we propose an efficient framework with provable guarantees for neural pruning, which is based on the existing theory of coresets such as (Braverman et al., 2016). Coresets decrease massive inputs to smaller instances while maintaining a good provable approximation of the original set with respect to a given function. Our main idea is to treat neurons of a neural network as inputs in a coreset framework. Specifically, we reduce the number of neurons in layer $i$ by constructing a coreset of neurons in this layer that provably approximates the output of neurons in layer $i + 1$ and discarding the rest. The coreset algorithm provides us with the choice of neurons in layer $i$ and with the new weights connecting these neurons to layer $i+1$. The coreset algorithm is applied layer-wise from the bottom to the top of the network.

The size of the coreset, and consequently the number of remaining neurons in layer $i$, is provably related to the approximation error of the output for every neuron in layer $i + 1$. Thus, we can theoretically derive the trade-off between the compression rate and the approximation error of any layer in the neural network. The coreset approximation of neurons provably holds for *any input*; thus our compression is data-independent.

Similar to our approach, Baykal et al. (2018) used coresets for model compression. However, their coresets are data-dependent; therefore, they cannot guarantee robustness over inputs. Moreover, they construct coresets of weights, while our approach constructs coresets of neurons. Neural pruning reduces the size of the weight tensors, while keeping the network dense. Hence the implementation of the pruned network requires no additional effort. Implementing networks with sparse weights (which is the result of weight pruning) is harder and in many cases does not result in actual computational savings.

Our empirical results on LeNet-300-100 for MNIST (LeCun et al., 1998) and VGG-16 (Simonyan & Zisserman, 2014) for CIFAR-10 (Krizhevsky, 2009) demonstrate that our framework based on coresets of neurons outperforms sampling-based coresets by improving compression without sacrificing the accuracy. Finally, our construction is very fast; it took about 56 sec. to compress each dense layer in the VGG-16 network using the platform specified in the experimental section.

**Our Contributions:**   We propose an efficient, data-independent neural pruning algorithm with a provable trade-off between the compression rate and the output approximation error. This is the first framework to perform neural pruning via coresets. We provide theoretical compression rates for some of the most popular neural activation functions summarized in Table 2.

## 2   RELATED WORK

### 2.1   CORESETS

Our compression algorithm is based on a data summarization approach known as coresets. Over the past decade, coreset constructions have been recognized for high achievements in data reduction in a variety of applications, including $k$-means, SVD, regression, low-rank approximation, PageRank, convex hull, and SVM; see details in Phillips (2016). Many of the non-deterministic coreset based

methods rely on the sensitivity framework, in which elements of the input are sampled according to their sensitivity (Langberg & Schulman, 2010; Braverman et al., 2016; Tolochinsky & Feldman, 2018), which is used as a measure of their importance. The sampled elements are usually reweighted afterwards.

## 2.2 MODEL COMPRESSION

State-of-the-art neural networks are often overparameterized, which causes a significant redundancy of weights. To reduce both computation time and memory requirements of trained networks, many approaches aim at removing this redundancy by model compression.

**Weight Pruning:** Weight pruning was considered as far back as 1990 (LeCun et al., 1990), but has recently seen more study (Lebedev & Lempitsky, 2016; Dong et al., 2017). One of the most popular approaches is pruning via sparsity. Sparsity can be enforced by $L_1$ regularization to push weights towards zero during training (Hu et al., 2016). However, it was observed (Han et al., 2015) that after fine-tuning of the pruned network, $L_2$ regularized network outperformed $L_1$, as there is no benefit to pushing values towards zero compared to pruning unimportant (small weight) connections.

The approach in Denton et al. (2014) exploits the linearity of the neural network by finding a low-rank approximation of the weights and keeping the accuracy within 1% of the uncompressed model. Jacob et al. (2018) performs quantization of the neural network's weights and suggests a new training procedure to preserve the model accuracy after the quantization.

These methods showed high compression rates, e.g., the compression rate of AlexNet can reach 35x with the combination of pruning, quantization, and Huffman coding (Han et al., 2016). Nevertheless, strong provable worst-case analysis is noticeably absent for most weight pruning methods.

**Neural pruning:** Weight pruning leads to an irregular network structure, which needs a special treatment to deal with sparse representations, making it hard to achieve actual computational savings. On the other hand, neural pruning (Hu et al., 2016) and filter pruning in CNNs (e.g, Zhuang et al. (2018); Li et al. (2017); Liu et al. (2017) simply reduce the size of the tensors.

The method in Hu et al. (2016) first identifies weak neurons by analyzing their activiations on a large validation dataset. Then those weak neurons are pruned and the network is retrained. The processes are repeated several times. Zhuang et al. (2018) introduces channel pruning based on the contribution to the discriminative power. These methods are data-dependent; thus they cannot provide guarantees of approximation error for any future input.

Li et al. (2017) measures the importance of channels by calculating the sum of absolute values of weights. Other channel pruning methods either impose channel-wise sparsity in training, followed by pruning channels with small scaling factors, and fine-tuning (e.g, Liu et al. (2017)) or perform channel pruning by minimizing the reconstruction error of feature maps between the pruned and pre-trained model (e.g., He et al. (2017).) These methods lack provable guarantees on the trade-offs between their accuracy and compression.

**Coreset-Based Model Compression** Similar to our work, the approach in Baykal et al. (2018) uses corests for model compression. However, they construct coresets of weights, while we construct coresets of neurons. Their approach computes the importance of each weight, which is termed sensitivity, using a subset from the validation set. The coreset is chosen for the specific distribution (of data) so consequently, the compressed model is data-dependent. In our construction, the input of the neural network is assumed to be an arbitrary vector in $\mathbb{R}^d$ and the sensitivity of a neuron is computed for every input in $\mathbb{R}^d$. This means that we create a data-independent coreset; its size is independent of the properties of the specific data at hand, and the compression provably approximates any future test sample.

Dubey et al. (2018) builds upon k-means coresets by adding a sparsity constraint. The weighting of the filters in the coreset is obtained based on their activation magnitudes over the training set. The compression pipeline also includes a pre-processing step that follows a simple heuristic that eliminates filters based on the mean of their activation norms over the training set. This construction is obviously data-dependent and it uses corsets as an alternative mechanism for low-rank approximation of filters.

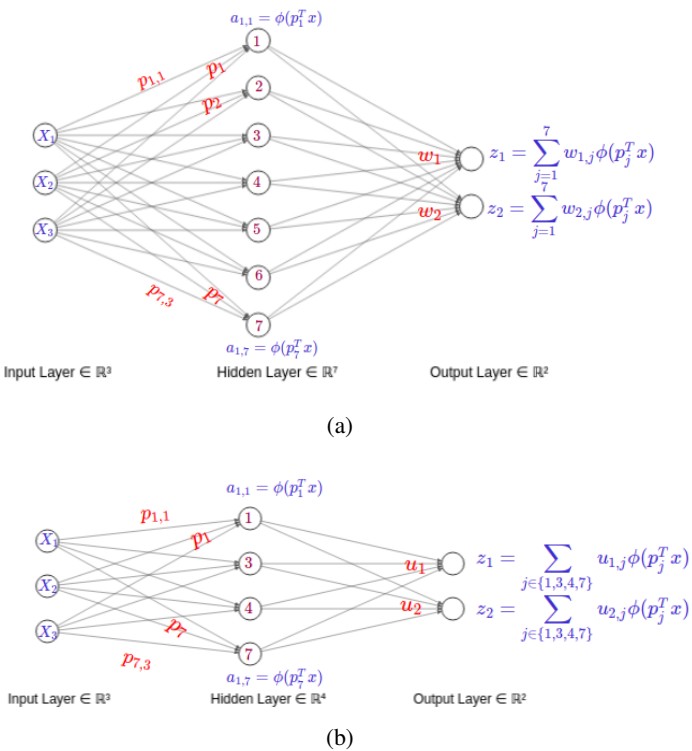

Figure 1: Illustration of our neuron coreset construction on a toy example: (a) a full network, (b) the compressed network. Both neurons in the second layer in (b) choose the same coreset comprising neurons $\{1, 2, 3, 7\}$ from layer 1, but with different weights. The compressed network has pruned neurons $\{2, 5, 6\}$ from layer 1.

## 3  METHOD

We propose an algorithm for compressing layer $i$ and we apply it to all layers from the bottom to the top of the network. We first give an intuitive description of the algorithm. We then formalize it and provide a theoretical analysis of the proposed construction.

### 3.1  DATA-INDEPENDENT CORESET FOR NEURAL PRUNING

Let $a_j^i = \phi(p_j^T x)$ be the $j$th neuron in layer $i$, in which $p_j$ denotes its weights, and $x$ denotes an arbitrary input in $\mathbb{R}^d$ (see Figure 1, top). We first consider a single neuron in layer $i + 1$. The linear part of this neuron is $z = \sum_{j=1}^{|P|} w(p_j)\phi(p_j^T x)$. We would like to approximate $z$ by $\tilde{z} = \sum_{l \in J^*} u(p_l)\phi(p_l^T x)$ where $J^* \subset \{1, ..., |P|\}$ is a small subset, and we want this approximation to be bounded by a multiplicative factor that holds for any $x \in \mathbb{R}^d$. Unfortunately, our result in Theorem 6 shows that this idealized goal is impossible. However, we show in Theorem 7 and Corollary 8 that we can construct a small coreset $C$, such that $|z - \tilde{z}| \leq \varepsilon$ for any input $x \in \mathbb{R}^d$.

Algorithm 1 summarizes the coreset construction for a single neuron with an activation function $\phi$, (our results for common neural activation functions are summarized in Table 2). Algorithm 2 and Corollary 9 show the construction of a single coreset with possibly different weights for all neurons in layer $i + 1$ (see Figure 1, bottom).

### 3.2  PRELIMINARIES

**Definition 1** (weighted set). *Let $P \subset \mathbb{R}^d$ be a finite set, and $w$ be a function that maps every $p \in P$ to a* weight *$w(p) > 0$. The pair $(P, w)$ is called a* weighted set.

---

**Algorithm 1:** CORESET$(P, w, m, \phi, \beta)$

---

**Input:**  A weighted set $(P, w)$,
          an integer (sample size) $m \geq 1$,
          an (activation) function $\phi : \mathbb{R} \to [0, \infty)$,
          an upper bound $\beta > 0$.

**Output:** A weighted set $(C, u)$; see Theorem 7 and Corollary 8.

1 **for** *every* $p \in P$ **do**

2 $\quad \mathrm{pr}(p) := \dfrac{w(p)\phi(\beta \|p\|)}{\sum_{q \in P} w(q)\phi(\beta \|q\|)}$

3 $\quad u(p) := 0$

4 $C \leftarrow \emptyset$

5 **for** $m$ *iterations* **do**

6 $\quad$ Sample a point $q$ from $P$ such that $p \in P$ is chosen with probability $\mathrm{pr}(p)$.

7 $\quad C := C \cup \{q\}$

8 $\quad u(q) := u(q) + \dfrac{w(q)}{m \cdot \mathrm{pr}(q)}$

9 **return** $(C, u)$

---

A coreset in this paper is applied on a query space which consists of an input weighted set, an objective function, and a class of models (queries) as follows.

**Definition 2** (Query space). *Let* $P' = (P, w)$ *be a weighted set called the* input set. *Let* $X \subseteq \mathbb{R}^d$ *be a set, and* $f : P \times X \to [0, \infty)$ *be a* loss function. *The tuple* $(P, w, X, f)$ *is called a* query space.

Given a set of points $P$ and a set of queries $X$, a coreset of $P$ is a weighted set of points that provides a good approximation to $P$ for any query $x \in X$. We state the definition of coresets with multiplicative guarantees below, though we shall also reference coresets with additive guarantees.

**Definition 3** ($\varepsilon$-coreset, multiplicative guarantee). *Let* $(P, w, X, f)$ *be a query space, and* $\varepsilon \in (0, 1)$ *be an error parameter. An* $\varepsilon$-coreset *of* $(P, w, X, f)$ *is a weighted set* $(Q, u)$ *such that for every* $x \in X$

$$\left| \sum_{p \in P} w(p)f(p, x) - \sum_{q \in Q} u(q)f(q, x) \right| \leq \varepsilon \sum_{p \in P} w(p)f(p, x)$$

The size of our coresets depends on two parameters: the complexity of the activation function which is defined below, and the sum of a supremum that is defined later. We now recall the well-known definition of VC dimension (Vapnik & Chervonenkis, 2015) using the variant from (Feldman & Langberg, 2011).

**Definition 4** (VC-dimension (Feldman & Langberg, 2011)). *Let* $(P, w, X, f)$ *be a query space. For every* $x \in \mathbb{R}^d$, *and* $r \geq 0$ *we define* $\mathrm{range}_{P,f}(x, r) := \{p \in P \mid f(p, x) \leq r\}$ *and* $\mathrm{ranges}(P, X, f) := \{C \cap \mathrm{range}_{P,f}(x, r) \mid C \subseteq P, x \in X, r \geq 0\}$. *For a set* ranges *of subsets of* $\mathbb{R}^d$, *the VC-dimension of* $(\mathbb{R}^d, \mathrm{ranges})$ *is the size* $|C|$ *of the largest subset* $C \subseteq \mathbb{R}^d$ *such that*

$$\left| \{C \cap \mathrm{range} \mid \mathrm{range} \in \mathrm{ranges}\} \right| = 2^{|C|}.$$

*The* VC-dimension of the query space $(P, X, f)$ *is the VC-dimension of* $(P, \mathrm{ranges}(P, X, f))$.

The VC-dimension of all the query spaces that correspond to the activation functions in Table 2 is $O(d)$, as most of the other common activation functions (Anthony & Bartlett, 2009).

The following theorem bounds the size of the coreset for a given query space and explains how to construct it. Unlike previous papers such as (Feldman & Langberg, 2011), we consider additive error and not multiplicative error.

**Theorem 5** (Braverman et al. (2016)). *Let* $d$ *be the VC-dimension of a query space* $(P, w, X, f)$. *Suppose* $s : P \to [0, \infty)$ *such that* $s(p) \geq w(p) \sup_{x \in X} f(p, x)$. *Let* $t = \sum_{p \in P} s(p)$, *and* $\varepsilon, \delta \in$

$(0, 1)$. *Let $c \geq 1$ be a sufficiently large constant that can be determined from the proof, and let $C$ be a sample (multi-set) of*

$$m \geq \frac{ct}{\varepsilon^2} \left( d \log t + \log \left( \frac{1}{\delta} \right) \right)$$

*i.i.d. points from $P$, where for every $p \in P$ and $q \in C$ we have $\mathrm{pr}(p = q) = s(p)/t$. Then, with probability at least $1 - \delta$,*

$$\forall x \in X : \left| \sum_{p \in P} w(p) f(p, x) - \sum_{q \in C} \frac{w(q)}{m \mathrm{pr}(q)} \cdot f(q, x) \right| \leq \varepsilon.$$

---

**Algorithm 2:** CORESET PER LAYER$(P, w_1, \cdots, w_k, m, \phi, \beta)$

---

| **Input:** | Weighted sets $(P, w_1), \cdots, (P, w_k)$, |
| | an integer (sample size) $m \geq 1$, |
| | an (activation) function $\phi : \mathbb{R} \to [0, \infty)$, |
| | an upper bound $\beta > 0$. |
| **Output:** | A weighted set $(C, u)$; see Theorem 7. |

1 **for** *every $p \in P$* **do**

2 $\quad \mathrm{pr}(p) := \dfrac{\max_{i \in [k]} w_i(p) \phi(\beta \|p\|)}{\sum_{q \in P} \max_{i \in [k]} w_i(q) \phi(\beta \|q\|)}$

3 $\quad u(p) := 0$

4 $C \leftarrow \emptyset$

5 **for** *$m$ iterations* **do**

6 $\quad$ Sample a point $q$ from $P$ such that $p \in P$ is chosen with probability $\mathrm{pr}(p)$.

7 $\quad C := C \cup \{q\}$

8 $\quad \forall i \in [k] : u_i(q) := u_i(q) + \dfrac{w_i(q)}{m \cdot \mathrm{pr}(q)}$

9 **return** $(C, u_1, \cdots, u_k)$

---

### 3.3 MAIN THEORETICAL RESULTS

Most of the coresets provide a $(1 + \varepsilon)$-multiplicative factor approximation for every query that is applied on the input set. The bound on the coreset size is independent or at least sub-linear in the original number $n$ of points, for any given input set. Unfortunately, the following theorem proves that it is impossible to compute small coresets for many common activation functions such as ReLU. This holds even if there are constraints on both the length of the input set and the test set of samples.

**Theorem 6** (No coreset for multiplicative error). *Let $\phi : \mathbb{R} \to [0, \infty)$ such that $\phi(b) > 0$ if and only if $b > 0$. Let $\alpha, \beta > 0$, $\varepsilon \in (0, 1)$ and $n \geq 1$ be an integer. Then there is a set $P \subseteq \mathbb{B}_\alpha(0)$ of $n$ points such that if a weighted set $(C, u)$ satisfies $C \subseteq P$ and*

$$\forall x \in \mathbb{B}_\beta(0) : \left| \sum_{p \in P} \phi(p^T x) - \sum_{q \in C} u(q) \phi(q^T x) \right| \leq \varepsilon \sum_{p \in P} \phi(p^T x), \tag{1}$$

*then $C = P$.*

The proof of Theorem 6 is provided in Appendix A.1.

The following theorem motivates the usage of additive $\varepsilon$-error instead of multiplicative $(1 + \varepsilon)$ error. Fortunately, in this case there is a bound on the coreset's size for appropriate sampling distributions.

**Theorem 7.** *Let $\alpha, \beta > 0$ and $(P, w, \mathbb{B}_\beta(0), f)$ be a query space of VC-dimension $d$ such that $P \subseteq \mathbb{B}_\alpha(0)$, the weights $w$ are non-negative, $f(p, x) = \phi(p^T x)$ and $\phi : \mathbb{R} \to [0, \infty)$ is a non-decreasing function. Let $\varepsilon, \delta \in (0, 1)$ and*

$$m \geq \frac{ct}{\varepsilon^2} \left( d \log t + \log \left( \frac{1}{\delta} \right) \right)$$

*where*

$$t = \phi(\alpha\beta) \sum_{p \in P} w(p)$$

*and $c$ is a sufficiently large constant that can be determined from the proof.*

*Let $(C, u)$ be the output of a call to* CORESET$(P, w, m, \phi, \beta)$*; see Algorithm 1. Then, $|C| \leq m$ and, with probability at least $1 - \delta$,*

$$\left| \sum_{p \in P} w(p)\phi(p^T x) - \sum_{p \in C} u(p)\phi(p^T x) \right| \leq \varepsilon.$$

The proof is provided in Appendix A.2.

As weights of a neural network can take positive and negative values, and the activation functions $\phi : \mathbb{R} \to \mathbb{R}$ may return negative values, we generalize our result to include negative weights and any monotonic (non-decreasing or non-increasing) bounded activation function in the following corollary.

**Corollary 8.** *Let $(P, w, \mathbb{B}_\beta(0), f)$ be a general query spaces, of VC-dimension $O(d)$ such that $f(p, x) = \phi(p^T x)$ for some monotonic function $\phi : \mathbb{R} \to \mathbb{R}$ and $P \subseteq \mathbb{B}_\alpha(0)$. Let*

$$s(p) = \sup_{x \in X} |w(p)\phi(p^T x)|$$

*for every $p \in P$. Let $c \geq 1$ be a sufficiently large constant that can be determined from the proof, $t = \sum_{p \in P} s(p)$, and*

$$m \geq \frac{ct}{\varepsilon^2} \left( d \log t + \log\left(\frac{1}{\delta}\right) \right).$$

*Let $(C, u)$ be the output of a call to* CORESET$(P, w, m, \phi, \beta)$*; see Algorithm 1. Then, $|C| \leq m$ and, with probability at least $1 - \delta$,*

$$\forall x \in \mathbb{B}_\beta(0) : \left| \sum_{p \in P} w(p)\phi(p^T x) - \sum_{p \in C} u(p)\phi(p^T x) \right| \leq \varepsilon.$$

The proof of Corollary 8 is provided in Appendix A.3.

### 3.4 FROM CORESET PER NEURON TO CORESET PER LAYER

Applying Algorithm 1 to each neuron in a layer $i + 1$ could result in the situation that a neuron in layer $i$ is selected to the coreset of some neurons in layer $i + 1$, but not to others. In this situation, it cannot be removed. To perform neuron pruning, every neuron in layer $i + 1$ should select the same neurons for its coreset, maybe with different weights. Thus, we wish to compute a single coreset for multiple weighted sets that are different only by their weight function. Each such a set represents a neuron in level $i + 1$, which includes $k$ neurons. Algorithm 2 and Corollary 9 show how to compute a single coreset for multiple weighted sets. Figure 1 provides an illustration of the layer pruning on a toy example.

**Corollary 9** (Coreset per Layer). *Let $(P, w_1, \mathbb{B}_\beta(0), f), \ldots, (P, w_k, \mathbb{B}_\beta(0), f)$ be $k$ query spaces, each of VC-dimension $O(d)$ such that $f(p, x) = \phi(p^T x)$ for some non-decreasing $\phi : \mathbb{R} \to [0, \infty)$ and $P \subseteq \mathbb{B}_\alpha(0)$. Let*

$$s(p) = \max_{i \in [k]} \sup_{x \in X} w_i(p)\phi(p^T x)$$

*for every $p \in P$. Let $c \geq 1$ be a sufficiently large constant that can be determined from the proof, $t = \sum_{p \in P} s(p)$*

$$m \geq \frac{ct}{\varepsilon^2} \left( d \log t + \log\left(\frac{1}{\delta}\right) \right).$$

*Let $(C, u_1, \cdots, u_k)$ be the output of a call to* CORESET$(P, w_1, \cdots, w_k, m, \phi, \beta)$*; see Algorithm 2. Then, $|C| \leq m$ and, with probability at least $1 - \delta$,*

$$\forall i \in [k], x \in \mathbb{B}_\beta(0) : \left| \sum_{p \in P} w_i(p)\phi(p^T x) - \sum_{p \in C} u_i(p)\phi(p^T x) \right| \leq \varepsilon.$$

| Network | Error(%) | # Parameters | Compression Ratio |
|---|---|---|---|
| LeNet-300-100 | 2.16 | 267K | |
| LeNet-300-100 Pruned | **2.03** | **26K** | **90%** |
| VGG-16 | 8.95 | 1.4M | |
| VGG-16 Pruned | **8.16** | **350K** | **75%** |

Table 1: Empirical evaluations of our coresets on existing architectures for MNIST and CIFAR-10. Note the improvement of accuracy in both cases!

The proof follows directly from the observation in Theorem 5 that $s(p) \geq w(p) \sup_{x \in X} f(p, x)$.

## 4 EXPERIMENTS

We first test our neural pruning with coresets on two popular models: LeNet-300-100 on MNIST (LeCun et al., 1998), and VGG-16 (Simonyan & Zisserman, 2014) on CIFAR-10 (Krizhevsky, 2009). We then compare the compression rate of our coreset (Neuron Coreset) to the compression methods based on the following sampling schemes:

**Baselines:** uniform sampling, percentile (which deterministically retains the inputs with the highest norms), and Singular Value Decomposition (SVD);

**Schemes for matrix sparsification:** based on L1 and L2 norms and their combination (Drineas & Zouzias, 2011; Achlioptas et al., 2013; Kundu & Drineas, 2014);

**Sensitivity sampling:** CoreNet and CoreNet++ (Baykal et al., 2018).

In all experiments we used ReLU networks and we computed the average error of the tested algorithms after performing each test ten times. For every layer, after applying neural pruning the remaining weights were fine-tuned until convergence. The experiments were implemented in PyTorch (Paszke et al., 2017) on a Linux Machine using an Intel Xeon, 32-core CPU with 3.2 GHz, 256 GB of RAM and Nvidia TitanX and Quadro M4000 GPUs . The source code of our method can be found at: `https://github.com/BenMussay/Data-Independent-Neural-Pruning-via-Coresets`.

### 4.1 COMPRESSING LENET AND VGG

LeNet-300-100 network comprises two fully connected hidden layers with 300 and 100 neurons correspondingly, trained on MNIST data set. Our coresets were able to prune roughly $90\%$ of the parameters and our compression did not have any associated accuracy cost – in fact, it slightly improved the classification accuracy.

VGG-16 (Simonyan & Zisserman, 2014) includes 5 blocks comprising convolutional and pooling layers, followed by 3 dense layers – the first two with 4096 neurons and the last with 1000 neurons. The model was trained and tested on CIFAR-10. We applied our algorithm for neural pruning to the dense layers, which have the largest number parameters. Our experiment showed slight improvement in accuracy of classification while the number of parameters decreased by roughly $75\%$. We summarize our findings in Table 1.

### 4.2 CORESETS ON RELU

We analyzed the empirical trade-off between the approximation error and the size of the coreset, constructed by Algorithm 1 and Corollary 8, in comparison to uniform sampling, which also implements Algorithm 1, but sets the probability of a point to $1/n$ ($n$ is the size of the full set), and to percentile, which deterministically retains the inputs with the highest norms (note that in percentile the points are not weighted). We ran three tests, varying the distribution of weights. In the first and second tests (Figure 2, (a) and (b)) the weights were drawn from the Gaussian and Uniform distributions respectively. The total number of neurons was set to 1000. We selected subsets of neurons

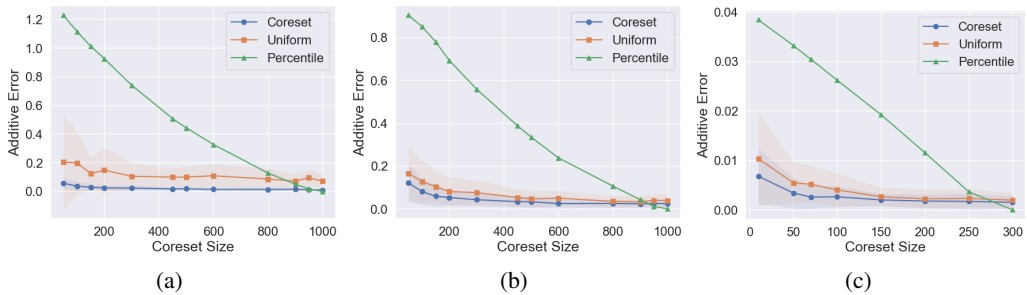

(a)                                    (b)                                    (c)

Figure 2: Approximation error of a single neuron on MNIST dataset across different coreset sizes. The weights of the points in (a) are drawn from the Gaussian distribution, in (b) from the Uniform distribution and in (c) we used the trained weights from LeNet-300-100. Our coreset, computed by Algorithm 1 and Corollary 8, outperforms other reduction methods.

of increasing sizes from 50 to 1000 with a step of 50. In the third test (Figure 2, (c)) we used the trained weights from the first layer of Lenet-300-100 including 300 neurons. We varied the coreset size from 50 to 300 with a step 50. To evaluate the approximation error, we used images from MNIST test set as queries. Each point in the plot was computed by 1) running the full network and the compressed network (with corresponding compression level) on each image $x$ in the test set, 2) computing additive approximation error $\left|\sum_{p \in P} w(p)\phi(p^T x) - \sum_{p \in C} u(p)\phi(p^T x)\right|$, 3) averaging the resulting error over the test set. In all three tests, our coresets outperformed the tested methods across all coreset sizes.

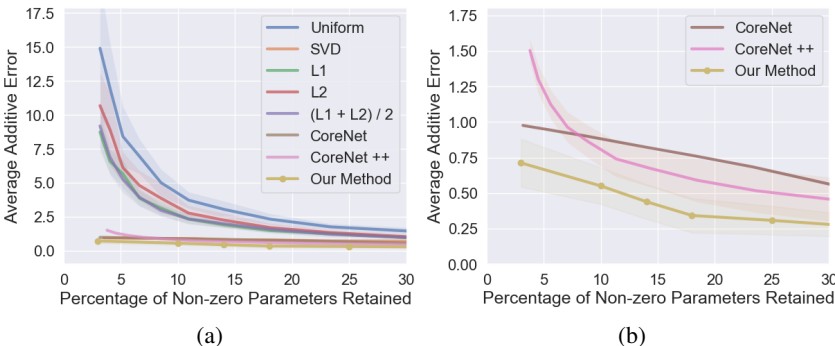

(a)                                    (b)

Figure 3: Average accuracy of various algorithms on LeNet-200-105 on MNIST dataset across different sparsity rates. Plot (a) shows the results of all tested methods. Plot (b) focuses on the three top methods. Our method, which constructs neural coresets by applying the Algorithm 2 and Corollary 8 in a layer-by-layer fashion, outperforms other coreset-based algorithms.

### 4.3 COMPARISON WITH OTHER METHODS.

We compare the average approximation error vs. compression rates of our neural pruning coreset with several other well-known algorithms (listed above). We run these tests on LeNet-200-105 architecture, trained and tested on MNIST, and we measure the corresponding average approximation error as defined in (Baykal et al., 2018):

$$error_{\mathcal{P}_{test}} = \frac{1}{\mathcal{P}_{test}} \sum_{x \in \mathcal{P}_t est} \|\phi_{\hat{\theta}}(x) - \phi_\theta(x)\|_1,$$

where $\phi_{\hat{\theta}}(x)$ and $\phi_\theta(x)$ are the outputs of the approximated and the original networks respectively.

The results are summarized in Figure 3. As expected, all algorithms perform better with lower compression, but our algorithm outperforms the other methods, especially for high compression rates.

### 4.4 ABLATION ANALYSIS

The proposed compression framework includes for every layer, a selection of neurons using Algorithm 2, followed by fine-tuning. We performed the following ablation analysis to evaluate the contribution of different parts of our framework on LeNet-300-100 trained on MNIST. First, we removed the fine-tuning, to test the improvement due to Algorithm 2 over the uniform sampling. Figure 4, (a) shows the classification accuracy without fine-tuning as a function of the compression rate. Figure 4, (b) shows that fine-tuning improves both methods, but the advantage of the coreset is still apparent across almost all compression rates and it increases at the higher compression rates. Note that the model selected by the coreset can be fine-tuned to 98% classification accuracy for any compression rate, while the model chosen uniformly cannot maintain the same accuracy for high compression rates.

These results demonstrate that our coreset algorithm provides better selection of neurons compared to uniform sampling. Moreover, it requires significantly less fine-tuning: fine-tuning until convergence of the uniform sampling took close to 2 epochs, while fine-tuning of our method required about half of that time.

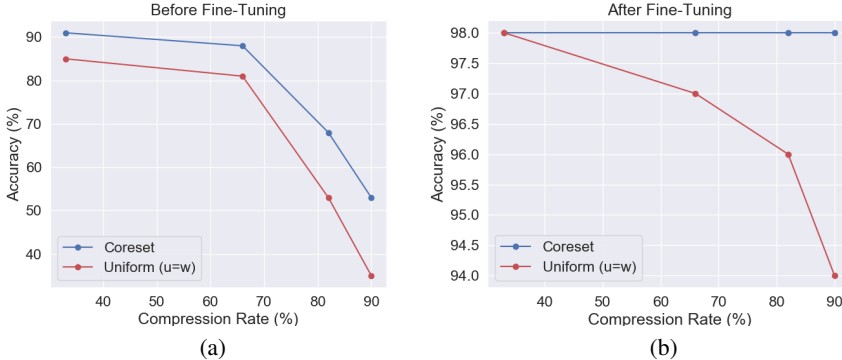

Figure 4: Average accuracy over 5 runs of the proposed framework and the uniform baseline on LeNet-300-100 on MNIST dataset across different compression rates. Plot (a) shows the results before fine-tuning, plot (b)-after fine-tuning. The fine-tuning was done until convergence. Fine-tuning of the uniform sampling almost doubles in time compared to the fine-tuning of the coreset.

## 5 CONCLUSION

We proposed the first neural pruning algorithm with provable trade-offs between the compression rate and the approximation error for any future test sample. We base our compression algorithm on the coreset framework and construct coresets for most common activation functions. Our tests on ReLU networks show high compression rates with no accuracy loss, and our theory guarantees the worst case accuracy vs. compression trade-off for any future test sample, even an adversarial one. In this paper we focused on pruning neurons. In future work, we plan to extend the proposed framework to pruning filers in CNNs, to composition of layers, and to other architectures.

### ACKNOWLEDGMENTS

We thank Rafi Dalla-Torre, Matan Weksler and Benjamin Lastmann from Samsung Research Israel for the fruitful debates and their technical support.

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

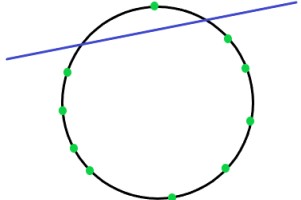 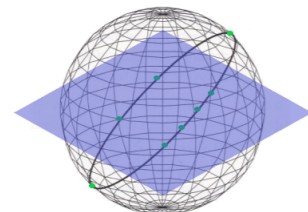

Figure 5: **(left)** Any point on a circle can be separated from the other points via a line. **(right)** The same holds for a circle which is the intersection of a $d$-dimensional sphere and a hyperplane; see Theorem 6.

# A APPENDIX

## A.1 PROOF OF THEOREM 6

Consider the points on $\mathbb{B}_\alpha(0)$ whose norm is $\alpha$ and last coordinate is $\alpha/2$. This is a $(d-1)$-dimensional sphere $S$ that is centered at $(0, \cdots, 0, \alpha/2)$. For every point $p$ on this sphere there is a hyperplane that passes through the origin and separates $p$ from the rest of the points in $S$. Formally, there is an arbitrarily short vector $x_p$ (which is orthogonal to this hyperplane) such that $x_p^T p > 0$, but $x_p^T q < 0$ for every $q \in S \setminus \{p\}$; see Fig. 5. By the definition of $\phi$, we also have $\phi(x_p^T p) > 0$, but $\phi(x_p^T q) = 0$ for every $q \in S \setminus \{p\}$.

Let $P$ be an arbitrary set of $n$ points in $S$, and $C \subset P$. Hence exists a point $p \in P \setminus C$. By the previous paragraph,

$$\left| \sum_{q \in P} \phi(x_p^T q) - \sum_{q \in C} u(q)\phi(x_p^T q) \right| = \left| \phi(x_p^T p) - 0 \right| = \phi(x_p^T p)$$

$$= \sum_{q \in P} \phi(x_p^T q) > \varepsilon \sum_{q \in P} \phi(x_p^T q).$$

Therefore $C$ does not satisfy equation 1 in Theorem 6.

## A.2 PROOF OF THEOREM 7

We want to apply Algorithm 1, and to this end we need to prove a bound that is independent of $x$ on the supremum $s$, the total supremum $t$, and the VC-dimension of the query space.

**Bound on $f(p, x)$.** Put $p \in P$ and $x \in \mathbb{B}_\beta(0)$. Hence,

$$f(p, x) = \phi(p^T x) \leq \phi(\|p\| \, \|x\|) \tag{2}$$
$$\leq \phi(\|p\| \, \beta) \tag{3}$$
$$\leq \phi(\alpha\beta), \tag{4}$$

where equation 2 holds by the Cauchy-Schwarz inequality and since $\phi$ is non-decreasing, equation 3 holds since $x \in \mathbb{B}_\beta(0)$, and equation 4 holds since $p \in \mathbb{B}_\alpha(0)$.

**Bound on the total sup $t$.** Using our bound on $f(p, x)$,

$$t = \sum_{p \in P} s(p) = \sum_{p \in P} w(p)\phi(\|p\| \, \beta) \leq \phi(\alpha\beta) \sum_{p \in P} w(p),$$

where the last inequality is by equation 4.

**Bound on the VC-dimension** of the query space $(P, w, \mathbb{B}_\beta(0), f)$ is $O(d)$ as proved e.g. in Anthony & Bartlett (2009).

**Putting all together.** By applying Theorem 1 with $X = \mathbb{B}_\beta(0)$, we obtain that, with probability at least $1 - \delta$,

$$\forall x \in \mathbb{B}_\beta(0) : \left| \sum_{p \in P} w(p) f(p, x) - \sum_{q \in C} u(q) f(q, x) \right| \leq \varepsilon.$$

Assume that the last equality indeed holds. Hence,

$$\forall x \in \mathbb{B}_\beta(0) : \left| \sum_{p \in P} w(p) \phi(p^T x) - \sum_{q \in C} u(q) \phi(q^T x) \right| \leq \varepsilon.$$

## A.3 PROOF OF COROLLARY 8

We assume that $\phi$ is a non-decreasing function. Otherwise, we apply the proof below for the non-decreasing function $\phi^* = -\phi$ and corresponding weight $w^*(p) = -w(p)$ for every $p \in P$. The correctness follows since $w(p)\phi(p^T x) = w^*(p)\phi^*(p^T x)$ for every $p \in P$.

Indeed, put $x \in \mathbb{B}_\beta(0)$, and $\phi$ non-decreasing. Hence,

$$\left| \sum_{p \in P} w(p)\phi(p^T x) - \sum_{p \in C} u(p)\phi(p^T x) \right| \tag{5}$$

$$\leq \left| \sum_{\substack{p \in P \\ w(p)\phi(p^T x) \geq 0}} w(p)\phi(p^T x) - \sum_{\substack{p \in C \\ u(p)\phi(p^T x) \geq 0}} u(p)\phi(p^T x) \right| + \left| \sum_{\substack{p \in P \\ w(p)\phi(p^T x) < 0}} w(p)\phi(p^T x) - \sum_{\substack{p \in C \\ u(p)\phi(p^T x) < 0}} u(p)\phi(p^T x) \right| \tag{6}$$

$$\leq \left| \sum_{\substack{p \in P \\ w(p)\phi(p^T x) \geq 0}} w(p)\phi(p^T x) - \sum_{\substack{p \in C \\ u(p)\phi(p^T x) \geq 0}} u(p)\phi(p^T x) \right| + \left| \sum_{\substack{p \in P \\ w(p)\phi(p^T x) < 0}} |w(p)\phi(p^T x)| - \sum_{\substack{p \in C \\ u(p)\phi(p^T x) < 0}} |u(p)\phi(p^T x)| \right| \tag{7}$$

Equation 6 is obtained by separating each sum into points with positive and negative weights and applying Cauchy-Schwarz inequality. Next, we bound points with positive and negative weights separately using Theorem 7.

## A.4 CORESET FOR DIFFERENT ACTIVATION FUNCTIONS

| Activation Function | Definition |
| --- | --- |
| ReLU | $\max(x, 0)$ |
| $\sigma$ | $\frac{1}{1+e^{-x}}$ |
| binary | $\begin{cases} 0 & \text{for } x < 0 \\ 1 & \text{for } x \geq 0 \end{cases}$ |
| $\zeta$ | $\ln(1 + e^x)$ |
| soft-clipping | $\frac{1}{\alpha} \log \frac{1+e^{\alpha x}}{1+e^{\alpha(x-1)}}$ |
| Gaussian | $e^{-x}$ |

Table 2: Examples of activation functions $\phi$ for which we can construct a coreset of size $O(\frac{\alpha\beta}{\varepsilon^2})$ that approximates $\frac{1}{|P|} \sum_{p \in P} \phi(p^T x)$ with $\varepsilon$-additive error.

