# OpenReview forum: "Data-Independent Neural Pruning via Coresets"
_ICLR.cc/2020/Conference — Accept (Poster)_

### Official Review · AnonReviewer1 · 2019-10-24
**Official Blind Review #1**

**Rating:** 3

**Review:**

The authors proposed a model compressing method based on coreset framework. The goal of the paper is to reduce the number of neurons. The basic idea is sampling the neurons on each layer with probability equal to the neuron's max share among all the outputs to the next layer, and updating the weights associated with the remained neurons. Another main contribution is the authors provided theoretical analysis to guarantee the accuracy vs compression trade-off for all possible inputs.
Pros:

The proposed method is easy to understand and seems to make sense.
The theoretical analysis seems strong.
The experiment results on two datasets show the proposed method achieved high compression rate and improvement of accuracy.


Cons:

Despite the theoretical guarantee, it is not as clear on the value of the proposed method in real world. I would be better to test on more datasets and networks to verify the effectiveness of the proposed compressing method, as it claimed to be data-independent.


Although the method achieved very good experiment results, its contribution to the high accuracy is unclear, since the networks were fine-tuned after the compression. So how do we exactly evaluate the accuracy vs compression trade-off when there is no such trade-off shown in the experiments?


Questions and suggestions:


In the Fig 2, it seems that the performances of the proposed method and the percentile-based method should be close to each other, and the uniform sampling method should be worse than them. However the results are opposite. If it was not incorrect labeling in the figure, it would be good to add some analyses about this result.

To solve the second point in "Cons", is it possible to show the accuracy of the compressed model without fine-tune? Or still fine-tune the model but simply set u(q) = w(q) in the line 8 algorithm 1?

**Experience Assessment:**

I do not know much about this area.

**Review Assessment: Checking Correctness Of Derivations And Theory:**

I assessed the sensibility of the derivations and theory.

**Review Assessment: Checking Correctness Of Experiments:**

I assessed the sensibility of the experiments.

**Review Assessment: Thoroughness In Paper Reading:**

I read the paper at least twice and used my best judgement in assessing the paper.

---

> ### Author Response · Authors · 2019-11-12
> **Reply to Review#1**
>
> We thank the reviewer for the useful comments. We hope that the
> clarifications below and the improvements we made in the new
> version of the paper would justify raising the score and giving us
> an opportunity to present our result in ICLR20.
>
>
> Our theoretical results show the approximation error vs compression
> trade-off which holds for any input in R^d. Since the proof holds
> for any vector in R^d, it holds for any dataset in particular. We
> do not put any constraints on the data or on the weights, except
> for a bound on the norm. The size of the coreset and thus the level
> of compression depends on the norm bound. The higher the norm, the
> larger the coreset (less compression). Best practices recommend
> normalizing the input of the network and use batch normalization,
> which normalizes inputs to hidden layers. This is done for most of
> the benchmark networks. Thus, we do not expect any dramatic change
> of results in the experiments on other data sets. We showed the
> results on two data sets with different properties: hand-written
> digits and natural images with objects. We do not have time to run
> an additional experiment during the discussion period, but we will
> include compression results on an additional domain, such as human
> faces, and more categories in the final version of the paper.
>
> The fine-tuning is part of the proposed method for compression.
> However, to show the contribution of our coreset framework, we
> added a new section "Ablation Analysis", in which we provide a
> comparison with uniform sampling with and without fine-tuning. It
> shows that before fine-tuning the coreset is huge advantage over
> uniform sampling. After fine-tuning, we still observe that the
> coreset selection is advantageous over uniform selection. Note that
> 4% in MNIST is a large difference in favour of our method.
>
> In Figure 2, the labels are correct. We believe that the details of
> uniform sampling algorithm would be helpful to clarify the results.
> Specifically, after sampling the neurons uniformly at random, their
> weights are updated the same way as it done in Alg 1, line 8, but
> with probability 1/n (n is the number of neurons in the layer).
> This is done to maintain the approximation error due to reduction
> in the number of neurons. The percentile method selects a
> predefined number of neurons with the largest norm of incoming
> weights, but it does not care about the weight of the neuron. We
> thank the reviewer for pointing out this misunderstanding. We added
> similar clarification in the new version of the paper.

---

### Official Review · AnonReviewer3 · 2019-10-27
**Official Blind Review #3**

**Rating:** 8

**Review:**

------ after reading authors' response ------
Thanks for the complete response and revision. The new Fig 4 is very nice, and this helps address my concerns. I'm more favorable of the paper now, changing from "Weak Accept" to "Accept".

Note a small typo in the displayed equation just above Fig 4: the sum is over $P_{test}$ but it's written $P_{t}est$ (and also using \text makes it look nicer, $P_{\text{test}}$)

------- original review -----
The paper proposes a neuron pruning technique that can compress an existing pre-trained neural net (though the experiments actually do additional unspecified "fine-tuning" training). It is motivated by the need to compress neural nets so they work on embedded devices (smart phones, etc.), and it is in contrast to most other techniques that prune at training, or prune after-training but prune weights not nodes. They argue convincingly that pruning weights is awkward, as one has to work with sparse matrices which are only actually effective for extreme sparsity levels. They also claim another big benefit is that theirs is the first with with (1) provable guarantees, and (2) is data-independent.

I have a some criticism of the paper, but before I get lost in the details, let me say that I like the overall paper. I think it's a clever idea, it's a useful topic, the authors show very good understanding of the coreset literature, and it has some nice theory.  The paper is also well-written and easy to understand, and the appendix is short enough that I actually read it.

However, I have at least two major comments:


(1) The theorems are nice, but with the exception of Thm 6 (which I like), they are simple applications of existing results. My main issue is that you have not provided an end-to-end bound. There are two things lacking:

(1a) Lack of dealing with several layers, e.g., composing your approximation error. With an additive error instead of a relative error, does composition cause a major problem? Seems like this could be an easy theorem.

(1b) Lack of a clear final statement bounding the overall error. This is somewhat trivial (if you have a single pruned layer), but it makes the assumptions more clear. In particular, you assume the input x has norm bounded by beta. In this sense, you have not provided a "data-independent" guarantee.  Since you do not have a relative error bound, the norm of x is important.  Yet this also exposes something that really confuses me: for the ReLU activation, with non-negative inputs, this is positive homogeneous, i.e., phi( beta ||p|| ) = beta phi( ||p|| ). If you look at step 2 in Algo 2, you see that the choice of beta does not affect the probabilities (if phi is ReLU or anything else with this property). Thus we can choose beta arbitrarily... and thus you have a fixed additive bound, for an arbitrarily large input, which seems impossible!

(2) Experiments were very promising, but I'm not convinced about the baselines.

(2a) The fine-tuning after pruning wasn't described so I don't know how much effect it had. It makes sense to do this, but it means that it is less clear if your results were due to your theorem.  Please show results with and without the fine-tuning, and describe the fine-tuning (how many epochs of training?)

(2b) You do some abstract experiments with random weights, to test your theorem, which is a nice somewhat direct test of your results (I assume here you are not fine-tuning, as it doesn't make sense, right?). Also in the abstract setup, you could test this as a function of depth, since I'm worried that your error guarantees get worse as a function of depth. The experimental setup was vague: what are the inputs x (from a ball, or sphere? uniform?), was this averaged over many runs? What was this network (you change size when you go to the LeNet-300-100), especially, what was the 100% number of samples (1000?)?.

(2c) For table 3, taking the LeNet for example, you have 90% compression and improved error. This is nice, but to really convince me, in addition to adding the results without the fine-tuning, I'd like to see what you get with uniform pruning (say, with 85% compression) with and without fine-tuning. I don't have a good "baseline" expectation here, so while your improved error with 90% compression seems like a fantastic result, I suspect that one might get similar results (with say 85% compression) with trivial subsampling.


Some minor comments:

-- abstract, "guarantees the accuracy of the function" and "... on MNIST while improving the accuracy."  These are 2 very different meaning of "accuracy", so please be more precise, e.g., a per-layer approximation error vs classification error on testing data.

-- First paragraph of intro, saying networks are limited to HPC environments is hyperbole. These networks might need to be trained in an HPC environment, but most can be deployed on laptops (not an HPC environment). A bigger issue is deploying them on a smartphone.  Adding some quantitative numbers would strengthen your case (e.g., size of typical neural nets, and size of RAM in a smartphone).  Note that training requires much more memory due to memory explosion in backpropagation, but this does not effect runtime/deployment.

-- middle of page 2, "is very fast, ..." the comma should be a semicolon to make it grammatically correct.  Page 3, near bottom, "corests" is a typo.

-- personally I dislike things like Table 1, as they feel too much like boasting. You've chosen the columns carefully so it doesn't feel that meaningful, and you've already stated these things in the text. But it's not my paper.

-- def 4 is very hard to parse. Are these subsets or strict subsets?

-- Notation B_alpha could be explained; I'm used to seeing B_alpha(0), and you can always just write \forall x with ||x||<= alpha to make it super clear.

-- Your theorems require phi to be non-decreasing, but intuitively you can clearly handle non-increasing, since the set of inputs x \in B_beta is invariant to sign changes. More generally, you could assume the existence of some 1D function psi(t) such that phi( t ) <= psi( |t| ). I don't know if there are many more common activation functions, but this could give a wider class.  The change to the proofs is trivial, since you just replace psi( |t| ) for phi( alpha beta).

-- If Corollary 9 "follows directly from Theorem 5", why didn't Thm 7 and Corollary 8 also follow directly? You mean, it follows, but using the same simple bounding tricks from the appendix as used for Thm 7 and Corollary 8, right?

-- Fig 3 shows very nice results

-- A.1 Proof of Thm 6, you could use \subsetneq (with amssymb package) to be more clear that it is a strict subset, since I find \subset vague since different authors have different conventions.

**Experience Assessment:**

I have read many papers in this area.

**Review Assessment: Checking Correctness Of Derivations And Theory:**

I assessed the sensibility of the derivations and theory.

**Review Assessment: Checking Correctness Of Experiments:**

I assessed the sensibility of the experiments.

**Review Assessment: Thoroughness In Paper Reading:**

I read the paper thoroughly.

---

> ### Author Response · Authors · 2019-11-12
> **Reply to Review#3**
>
> We want to thank the reviewer for the detailed review and very
> helpful suggestions. We hope that with these improvements, the
> reviewer would consider increasing the score.
>
> 1. Indeed, Theorem 6 and 7 are our main theoretical results, while
> the other are corollaries as stated. Based on these theoretical
> results, we propose the first applications of coresets for neural
> pruning with very good experimental results.
>
> 1a. Approximating the entire network even for a simpler case,
> addressed in Baykal et al., 2018, results in a large error, thus we
> took a layer by layer approach: For every layer, after applying
> neural pruning the remaining weights are fine-tuned until
> convergence. We do not approximate the whole network and thus there
> is no need to bound this approximation. Deriving a  coreset for approximating
> a composition of layers is part of the future work.
> We have clarified the layer-wise approach in the new version of the paper.
>
> 1b. We agree that the probability of sampling a point does not
> depend on beta. However, the size of the coreset depends on the
> total sensitivity, which depends on beta. It is true that the
> approximation error does not depend directly on the norm of the
> query, but it depends on the size of the coreset (which depends on
> beta). For an arbitrary large beta, the size of the coreset will
> reach the size of the full data, but it will not change the
> approximation error. Best practices recommend normalizing the input
> of the network and use batch normalization, which normalizes the
> input to other layers. Thus, the value of beta in real data sets is
> not arbitrary large, which explains small coresets in our
> experiments.
>
> 2a. We have added the details about fine-tuning to the paper: we
> fine-tune until convergence, but coreset converges in half of the
> convergence time of the model chosen by the uniform sampling. The
> results of compression before fine-tuning are provided in new
> Section 4.4.
>
> 2b. The results in Figure 2 are provided without fine-tuning. We
> added the experimental setup to the paper.
>
> The experimental setting for showing the approximation error as a
> function of depth is not clear: First, each layer has several
> neurons and each has its own approximation error, thus it is not
> clear how to combine these errors as a representative error of the
> layer. Second, each layer might have different scaling of neurons'
> output. Thus, showing the approximation error in each layer without
> proper normalization (which is not trivial to define) will not
> provide the information about the error stability vs. depth. We
> feel that a proper experiment showing the effect of depth requires
> further consideration and cannot be finished within the discussion
> period. However, we will think about the experimental setting that
> would show the approximation error vs. depth for the final version
> of the paper.
>
> 2c.The fine-tuning is part of the proposed method for compression.
> However, to show the contribution of our coreset framework, we
> added a new section "Ablation Analysis", in which we provide a
> comparison with uniform sampling with and without fine-tuning. It
> shows that before fine-tuning the coreset has huge advantage over
> uniform sampling. After fine-tuning, we still observe that the
> coreset selection is advantageous over uniform selection. Note that
> 4% in MNIST is a large difference in favor of our method.
>
> "Minor Comments": Below, we provide our answers to some of the
> question. We amended the paper according to reviewer's suggestions.
>
> Q: def 4 is very hard to parse.
> A: Due to space limitation, we
> summarized a number of concepts in one definition, but we added
> references to classic book in PAC-learning for helpful intuition
> and discussion.
>
> Q: Are these subsets or strict subsets?
> A: Subsets, as stated.
> Otherwise the required 2^|C| subsets (all subsets of C) in
> Definition 4 would never hold.
>
> Q: Your theorems require phi to be non-decreasing, but intuitively
> you can clearly handle non-increasing. I don't know if there are
> many more common activation functions, but this could give a wider
> class.
> A: This is a nice observation! We have change Corollary 8 to
> extended the results from Theorem 7 to negative weights and
> monotonic function with negative values.
>
> Q: why didn't Thm 7 and Corollary 8 also follow directly from
> Theorem 5?
> A: Theorem 7 follows directly from Theorem 5, after
> bounding s and t. Indeed, this is the essence of its proof.
> Corollary 8 generalizes Theorem 5 to support non-positive weights
> (and as suggested by the reviewers, non-positive activation
> functions), which requires a proof.
>
> Q: Corollary 9 follows using the same simple bounding tricks from
> the appendix as used for Thm 7 and Corollary 8, right? A: Exactly.
> We added this clarification in the text.

---

### Official Review · AnonReviewer2 · 2019-10-28
**Official Blind Review #2**

**Rating:** 6

**Review:**

This paper provides a data-independent way for pruning neutrons in deep neural networks with a provable trade-off between its compression rate and the approximation error. The output of a layer of neurons is approximated by a corset of neurons in its preceding layer.

The pruning of neurons based on the coresets is shown to be effective when compared with other methods. The authors have validated it on two convolutional network architectures.

The paper starts from defining the coresets and introducing the VC dimension, and extends the theorem to more generalized cases.

The coreset seems to require the activation function to be non-negative, which will possibly limit the scope of application of the proposed theory.

**Experience Assessment:**

I do not know much about this area.

**Review Assessment: Checking Correctness Of Derivations And Theory:**

I did not assess the derivations or theory.

**Review Assessment: Checking Correctness Of Experiments:**

I assessed the sensibility of the experiments.

**Review Assessment: Thoroughness In Paper Reading:**

I read the paper at least twice and used my best judgement in assessing the paper.

---

> ### Author Response · Authors · 2019-11-12
> **Reply to Review#2**
>
> We thank the reviewer for very helpful comments, that inspired us
> to generalize the results to a broader family of activation
> functions, which include negative values. Specifically, we have
> updated Corollary 8 to include both negative weights and arbitrary
> bounded functions. We hope that these improvement justifies a
> higher score.

---

### Decision · Program_Chairs · 2019-12-19

**Decision:**

Accept (Poster)

**Comment:**

The rebuttal period influenced R1 to raise their rating of the paper.
The most negative reviewer did not respond to the author response.
This work proposes an interesting approach that will be of interest to the community.
The AC recommends acceptance.